# Conserved Sequences in the 5′ and 3′ Untranslated Regions of Jingmenvirus Group Representatives

**DOI:** 10.3390/v15040971

**Published:** 2023-04-15

**Authors:** Alexander G. Litov, Egor V. Okhezin, Ivan S. Kholodilov, Oxana A. Belova, Galina G. Karganova

**Affiliations:** 1Laboratory of Biology of Arboviruses, FSASI “Chumakov Federal Scientific Center for Research and Development of Immune-and-Biological Products of RAS” (Institute of Poliomyelitis), 108819 Moscow, Russia; 2Department of Biology, Lomonosov Moscow State University, 119234 Moscow, Russia

**Keywords:** Alongshan virus, Yanggou tick virus, Jingmenvirus group, flavi-like virus, segmented virus, untranslated region, RNA structure, 5′ UTR, 3′ UTR, tetraloop

## Abstract

The Jingmenvirus group (JVG), with members such as Jingmen tick virus (JMTV), Alongshan virus (ALSV), Yanggou tick virus (YGTV), and Takachi virus (TAKV), is drawing attention due to evidence of it causing disease in humans and its unique genome architecture. In the current work, complete untranslated regions (UTRs) of four strains of ALSV and eight strains of YGTV were obtained. An analysis of these sequences, as well as JVG sequences from GenBank, uncovered several regions within viral UTRs that were highly conserved for all the segments and viruses. Bioinformatics predictions suggested that the UTRs of all the segments of YGTV, ALSV, and JMTV could form similar RNA structures. The most notable feature of these structures was a stable stem-loop with one (5′ UTR) or two (3′ UTR) AAGU tetraloops on the end of a hairpin.

## 1. Introduction

Recently, many new viruses have been discovered, uncovering new viral groups and models of genome organization [1,2,3,4,5]. One such recently discovered group is segmented flavi-like viruses [3,6,7,8,9]. Phylogenetically segmented flavi-like viruses are divided in two distinct clades: the Guaico Culex virus group and the Jingmenvirus group (JVG) [10]. The JVG includes several members, such as Jingmen tick virus (JMTV), Alongshan virus (ALSV), Yanggou tick virus (YGTV), and Takachi virus (TAKV) [3,6,7,8,9]. JVG members are believed to be arboviruses [9,11] and have been detected in various arthropods, including ticks [3,7,8,12,13,14,15,16,17,18], mosquitoes [3,7], reptiles [19], and mammals (including humans) [7,16,17,18,20]. Moreover, there is evidence that JMTV and ALSV may be able to cause disease in humans [7,17].

JVG members are characterized by a segmented (+) RNA genome consisting of four segments. The first segment is monocistronic and encodes flavivirus-like RNA-dependent RNA polymerase. The second segment is polycistronic and encodes from two to three open reading frames (ORFs), including nuORF and VP1 (divided into VP1a and VP1b in ALSV, YGTV, and TAKV) [3,6]. The VP1 protein is believed to be a viral glycoprotein. Recent bioinformatic research has predicted that it has structural features of class-II viral fusion proteins, including a three-domain architecture with an ectodomain and a fusion peptide with interfacial hydrophobicity [21]. The third segment is also monocistronic and encodes a flavivirus-like helicase. The fourth segment encodes two ORFs, one of which is believed to encode capsid protein [3]. In contrast with classical flaviviruses [22] and members of the Guaico Culex virus group [23], every JVG segment is polyadenylated [3].

For RNA viruses, untranslated regions (UTRs) play a crucial role in the virus replication cycle, generally by playing a role in translation and replication, as well as in some other functions (dependent on the virus group) [24,25,26,27,28]. Usually, these functions are carried out by RNA interactions and are conserved within a virus group.

UTR conservation is just as important for segmented (−) RNA viruses. The genomes of bunyaviruses and arenaviruses exhibit complementary sequences in 5′ and 3′ UTRs of their genome segments, which were predicted to form panhandle structures [25]. In Bunyamwera virus, 11 first nucleotides at both the 3′ and 5′ UTRs are conserved within the virus for all segments [26]. The UTR of the influenza A virus has motif sequences, which are highly conserved among viral strains and among the eight genome segments, located in both the 3′ and 5′ UTRs. Additionally, there are segment-specific noncoding regions, of which the length and sequences are specific to each segment and viral species [27].

In all avian reoviruses, the first seven nucleotides at the 5′ end and the last five nucleotides at the 3′ end are conserved. It has been suggested that they may be involved in the transcription, replication, and encapsidation of viral transcripts [28]. The role of UTRs in the flavivirus life cycle has been extensively studied. Different UTR elements play roles in replication, translation, genome cyclization, life cycle regulation, RNA stability, and subgenomic flavivirus RNA (sfRNA) generation [24,29,30].

At the same time, the role of UTRs in the JVG is relatively unstudied. During its first isolation, JMTV was reported to conserve all segments of the 5′-GCAAGUGCA-3′ nucleotide in the 5′ UTR and the 5′-GGCAAGUGC and 5′-CAAGUG-3′ nucleotides in the 3′ UTR. This fact was interpreted as evidence that all four segments belonged to the same virus [3]. Later, when ALSV was isolated, only conserved nucleotides in the 5′ UTR were found [7]. In this work, we determine the sequences of the UTRs of four ALSV strains and eight YGTV strains. Using these data, as well as data found in the GenBank database, we investigate sequences and possible RNA structure conservation in the members of the JVG.

## 2. Materials and Methods

### 2.1. Viruses and Cell Cultures

Three previously isolated ALSV strains (Miass527, Miass502, and Miass519) [6,8] and eight YGTV strains (Bredy15-T22208, Bredy15-T22181, Bredy15-T22188, Bredy15-T22189, Plast15-T22436, Plast15-22438, Erzhin14-T20074, and Kartaly14-T19309) [8,13] were used in this work. Amplicons detected previously (Miass523 and Kursh17-T25456) were isolated during this work.

HAE/CTVM8 [31] and IRE/CTVM19 [32] tick cell cultures (kindly provided by Lesley Bell-Sakyi of the Tick Cell Biobank) were cultivated as described previously [6,8,13,31,32].

### 2.2. Obtaining of Virus-Containing Material

Previously isolated strains were cultivated on the HAE/CTVM8 cell line or the IRE/CTVM19 cell line for a period ranging from six months to three years, as described earlier [6,8,13]. The Miass523 and Kursh17-T25456 ALSV strains were isolated via persistence for two months in the IRE/CTVM19 [32] and HAE/CTVM8 [31] cell lines, respectively, using previously described methodology [6,13].

Cultural supernatants were screened to estimate viral load. Virus amount was assessed by band intensity after PCR and agarose gel electrophoresis using virus-specific oligonucleotides (Appendix A). If the estimated viral load was high, viral RNA was isolated directly from the supernate. If the virus load was low, virus-containing material was concentrated by ultracentrifugation, as described earlier [6]. Briefly, 15–35 mL of cultural supernate was centrifuged at 10,000 rpm at 10 °C for 30 min with an Optima L-90K Ultracentrifuge (Beckman Coulter, Brea, CA, USA) using an SW-28 rotor. The centrifuged supernate was transferred into fresh tubes and ultracentrifuged at 25,000 rpm at 10 °C for 6 h using the same centrifuge and rotor. Ultracentrifugation pellets were dissolved in phosphate buffer saline (Sigma, St. Loise, MO, USA) and further used for RNA isolation.

### 2.3. Rapid Amplification of cDNA Ends

Viral RNA was isolated from virus-containing material using TRI Reagent LS according to the manufacturer’s instructions. Rapid amplification of the cDNA ends (RACE) was performed using a Mint RACE cDNA amplification set (Evrogen, Moscow, Russia) based on the step-out RACE method [33].

Briefly, first strand synthesis was performed with viral RNA using PlugOligo (Evrogen, Moscow, Russia) as a 5′ adapter and a 3′ primer (Evrogen, Moscow, Russia) targeting polyA sequences with the Mint reverse transcriptase (Evrogen, Moscow, Russia) at 42 °C for 30 min. After that, IP-mix (Evrogen, Moscow, Russia) was added, and the probe was incubated at 42 °C for 1.5 h. A 5′ RACE PCR was performed with segment-specific oligonucleotides (Appendix A) and Step-out primer mix 1 (Evrogen, Moscow, Russia) using Encyclo polymerase (Evrogen, Moscow, Russia) according to the instructions in the Mint RACE cDNA amplification set manual (Evrogen, Moscow, Russia). A 3′ RACE PCR was performed using PCR Step-out primer mix 1 (Evrogen, Moscow, Russia) and segment-specific oligonucleotides (Appendix A) with Encyclo polymerase (Evrogen, Moscow, Russia) according to the manufacturer’s instructions.

### 2.4. Cloning of PCR Products

If the PCR product was too heterogeneous to be sequenced directly, it was cloned into the pCR 2.1 vector using The Original TA Cloning Kit (Invitrogen, Carlsbad, CA, USA) according to the manufacturer’s instructions. *E. coli* cells of the TOP10 strain were transformed with ligation products, and positive clones were blue-white selected, as per the instructions in the cloning kit manual. Plasmids were purified using a GeneJET Plasmid Miniprep Kit (Thermo Fisher Scientific, Roskilde, Denmark). The amounts of clones taken and sequences obtained for each sequence are presented in Appendix A.

### 2.5. Sequencing

Obtained PCR products were sequenced directly from gene-specific oligonucleotides (Appendix A). Plasmids were sequenced directly using M13 (-21) Primer (Applied Biosystems, Foster City, CA, USA) and custom M13_long_F primers (Appendix A). Sequencing was carried out using ABI PRISM BigDye™ Terminator v. 3.1 with an ABI PRISM 3500 instrument (Applied Biosystems, Foster City, CA, USA). Sequences were assembled using SeqMan v.7.0.0 software (DNAstar, Madison, WI, USA). In cases where several clones were taken, after the assembly of individual clones, assembled sequences were aligned in SeqMan v.7.0.0, and consensus sequences were extracted using the “Save consensus” function with default program parameters.

All obtained sequences were deposited in GenBank. Entries ON448346–ON448374, OQ789403-OQ789442, and OQ789396-OQ789399 were deposited, and entries: MN648770.2–MN648777.2, MW525314.2–MW525317.2 and MW525322.2–MW525325.2 were updated.

### 2.6. GenBank Dataset Preparation

Nucleotide sequences of JMTV were extracted from the GenBank database on 7 February 2023 using “Jingmen tick virus“ as the primary organism modifier and a length range of 2000–4000 nt. In total, 492 sequences were obtained. Obtained sequences were further filtered out, using the prototype JMTV SY84 strain sequences of individual segments. Sequences in the set with less than 70% homology with the prototype strain were discarded. This resulted in 4 alignments (1 for each JMTV segment), with 438 sequences in total. Each segment alignment was then individually aligned using ClustalW, and 3′ and 5′ UTRs were extracted.

Nucleotide sequences of ALSV were extracted from the GenBank database on 7 February 2023 using “Alongshan virus“ as the primary organism modifier and a length range of 2000–4000 nt. Obtained sequences were further filtered using the Miass527 strain sequences of individual segments. Sequences in the set with less than 70% homology with the prototype strain were discarded. Sequences deposited by our group were also filtered out because they either were already in the structural alignment or were not sequenced using the RACE approach and thus were unreliable. Each segment alignment was then individually aligned using ClustalW, and 3′ and 5′ UTRs were extracted. The 5′ UTR sequences of the ALSV strain NE-TH4 (ON408068 and ON408067) were also discarded because their first ~50 nt showed extreme homology to the 5S and 5.8S ribosomal RNA of ticks.

Nucleotide sequences of YGTV were extracted from the GenBank database on 27 February 2023 using “Yanggou tick virus“ as the primary organism modifier and a length range of 2000–4000 nt. Filtering was performed similarly to that described above for ALSV, with the Erzhin14-T20074 strain used as a reference.

Nucleotide sequences of TAKV were extracted from the GenBank database on 27 February 2023 using “Takachi virus“ as the primary organism modifier and a length range of 2000–4000 nt. No additional filtering was performed for TAKV.

Nucleotide sequences of GCXV were extracted from the GenBank database on 8 April 2023 using “Guaico Culex virus“ as the search term. Obtained sequences were further filtered using the LO35 strain sequences of individual segments. Sequences containing truncated UTR were discarded. For the construction of the 3′ UTR structure, only segment 1–4 sequences were used.

Nucleotide sequences of TBEV were extracted from the GenBank database on 8 April 2023 using “tick-borne encephalitis virus“ as the primary organism modifier and a length range of 10,000–12,000 nt. Obtained sequences were further filtered using the tick-borne encephalitis virus NCBI reference sequence (NC_001672.1). Sequences containing truncated UTR were discarded. For the 3′ UTR structure construction, terminal 109 genome nucleotides were used.

### 2.7. Bioinformatics and Visualization

LocARNA v. 1.9.2.0 [34] with default parameters was used to simultaneously predict RNA structures and align sequences. Obtained structural alignments were visualized as RNA structures using RNAplot from the ViennaRNA package [35], with the use of the “most informative sequence” and “annotate covariance of base pairs” modes and an RNApuzzler plotting layout algorithm.

For sequence logo visualization, WebLogo 3.7.12 was used, with the “ignore lowercase” option, “Classic (NA)” color scheme, no version fingerprint option, and other options as default [36].

Additionally, we performed an alternative analysis by first aligning sequences using ClustalW [37] and then using RNALalifold from the ViennaRNA package [35].

All image post-processing was performed with GIMP v. 2.10.28.

## 3. Results

### 3.1. 5′ UTR Sequences and Structures

In this work, we determined the sequences of the 5′ UTRs of four ALSV strains and eight YGTV strains. A visual examination of the sequences of all the segments showed striking similarities, and it was reported earlier that there were some homologous sequences between different segments in JMTV [3], so we merged the sequences of all four segments for both ALSV and YGTV into a single batch. Aligning these sequences using LocARNA (Figure 1) and ClustalW (Appendix A) produced similar results. The first six nucleotides of all the segments for both ALSV and YGTV had a 100% conserved 5′-AGUUAA-3′ sequence. Nucleotides 28–36 (5′-GCAAGUGCA-3′), G(40), and 60–62 (5′-GAC-3′) were also 100% conserved in both alignments.

We examined the ALSV and YGTV sequences from GenBank (Appendix A). Most of them showed that all features that were conserved in our sequences were present in them, except the first six nucleotides, which varied depending on the strain and segment. None of these conserved sequences were found in the TAKV 5′ UTR, but it is specifically noted by the authors that RACE was not performed for 5′ UTR [9] and thus 5′ ends might be lacking.

We used the obtained alignments to predict possible RNA structures (Figure 1B and Appendix A) with generally similar results. The predictions show that the 5′ UTRs of ALSV and YGTV formed a stem-loop structure, where conserved nucleotides 28–35 formed two GC pairs that supported an AAGU tetraloop. Conserved G(40) formed a GC or GU pair that supported the formation of the A—AWCR budge. According to the LocARNA predictions, the following stem structure varied in the different segments (Figure 1C–F) but generally seemed to involve seven consecutive nucleotide pairs in the formation. The LocARNA analysis of all the segments predicted the existence of an additional stem-loop structure close to the start codon region; however, it was much less conserved, and the prediction results varied for different segments.

Interestingly, JMTV was reported to have a conserved nucleotide sequence of 5′-GCAAGUGCA-3′ among all the segments in the 5′ UTRs [3]. In order to estimate the conservation of the abovementioned conserved elements, we extracted JMTV sequences from GenBank. The LocARNA alignment of all the segments showed little homology in the 5′ UTRs (Figure 2C). A visual examination of the alignment showed that the program seemed to be unable to properly align segment 2 sequences with the sequences of other segments, probably because of high differences in UTR lengths. We performed a LocARNA alignment separately for segment 2 (Figure 2A,D) and for segments 1, 3, and 4 (Figure 2B,E). According to these alignments, all the segments had a conserved 5′-GCAAGUGC-3′ element. This element seemed to form a similar structure, where two GC pairs supported an AAGU tetraloop. All the other elements found in ALSV and YGTV were not found in JMTV.

In many JMTV genomes, the first nucleotides of the genome were 5′-GUUAA-3′ (similarly to the 5′-AGUUAA-3′ we found in ALSV and YGTV). However, several strains had substitutions or insertions in this sequence. In the particular case of JMTV isolate TTP-Pool-3b segment 4, the UTR had 48 additional terminal 5′ UTR nucleotides.

Overall, our data show that the 5′ UTRs of JMTV, ALSV, and YGTV contained a conserved 5′-GCAAGUGC-3′ sequence in all the segments, which is likely to be involved in the formation of the RNA structure, where two GC pairs supported an AAGU tetraloop.

### 3.2. 3′ UTR Sequence and Structures

In this work, we determined the sequences of the 3′ UTRs for five ALSV strains and eight YGTV strains. Using the obtained data, an analysis of sequence conservation and RNA structures was performed, similar to that of the 5′ UTRs, resulting in the alignment of all four segments of both ALSV and YGTV (Figure 3 and Appendix A).

The first notable feature of the alignment was the appearance of the A-rich region at positions 282–298. This region was present in all the sequences, with lengths varying with virus, strain, and segment. It was the most pronounced in the YGTV segment 2 and 3 sequences, where more than 80% of A in the 30 nt sequences was observed. In several cases, we were unable to obtain a direct sequence from the RACE PCR products after this region. A bacterial-cloning approach showed that around 33% of the clones (see Appendix A) had truncated 3′ UTRs that ended at the A-rich region. In the case of Miass502 ALSV, we were unable to obtain a full 3′ UTR after 16 clone pickups.

Both LocARNA and ClustalW revealed two identical 100% conserved sequences within the 3′ UTRs: 5′-CAAGUG-3′ (positions 354–359 and 372–377; Figure 3A). The RNA structure prediction suggested that both these regions take part in the formation of a dumbbell-like structure. In both cases, conserved nucleotides formed a GC pair that supported an AAGU tetraloop. In the first case, the tetraloop was supported by two to three nucleotide pairs, and in the second, it was supported by seven to eight nucleotide pairs. Alignment of all the segments showed little support, but alignment of the individual segments predicted that the dumbbell-like structure might be located on the long RNA stem. An examination of the ALSV and TAKV sequences from GenBank showed that all the sequences that had full, non-shortened 3′ UTR sequences had both 5′-CAAGUG-3′ occurrences in them (Appendix A). In the case of YGTV, one GenBank sequence (MH688534) had a C→A mutation in the second region (Appendix A), with all the other sequences having both 5′-CAAGUG-3′ occurrences as absolutely conserved.

We performed an analysis of the JMTV sequences containing full 3′ UTRs (181 sequences) with LocARNA (Figure 4). The obtained alignment showed that the same two conserved regions could be identified within JMTV 3′ UTRs. The JMTV genome contained a polyA-rich region (243–263) that varied in length depending on the strain and segment. There were two highly conserved regions: 309–317 5′-GGCAAGUGC-3′ and 330–335 5′-CAAGUG-3′ (Figure 4A). The first one was conserved in all of the sequences, while the second one was different in two sequences. The JMTV isolate JTMV_3 segment 4 (MH155897 [18]) preserved GC pairs but had an AAGU→UCGG substitution. The JMTV isolate WS3 segment 3 (OM459839) had a G(359)→U substitution.

Similar to ALSV and YGTV, these conserved regions were predicted to be folded into a dumbbell-like structure with two AAGU tetraloops (Figure 4B). The first tetraloop was supported by three nucleotide pairs. The second tetraloop seemed to have a longer stem, similar to ALSV and YGTV, but apart from the first two pairs, it was not well-supported.

Overall, our data show that the 3′ UTRs of JMTV, ALSV, TAKV, and YGTV contained two conserved 5′-CAAGUG-3′ sequences in all the segments likely to be involved in the formation of RNA structures.

## 4. Discussion

We performed a thorough analysis of the conserved sequences in the JVG UTR regions based on those determined for ALSV and YGTV, as well as for all the sequences found in GenBank. Our research showed several highly conserved regions within the UTRs, as well as points of interest for future research.

Previously, when JMTV was discovered, researchers pointed out there was some conservation among all the segments of the JMTV SY84 strain [3]. According to our data, there were three highly conserved regions: 5′-GCAAGUGC-3′ in the 5′ UTRs and two 5′-CAAGUG-3′ elements in the 3′ UTRs of the JVG. These conserved sequences were likely to participate in the formation of RNA structures, where an AAGU tetraloop was supported by at least two GC pairs. This UTR pattern differs greatly from the UTR patterns reported for Guaico Culex virus (GCXV), the only member of the Guaico Culex virus group with well-studied molecular biology. It is reported that all the segments and isolates of GCXV have conserved 5′ UTR (5′-AAAUUAAAA-3′) and 3′ UTR (5′-CCCAUUU-3′) terminal sequences. The other highly conserved motif was found in the 3′ UTR (5′-AAWUAC-3′) of segments 1–4 and was predicted to form a loop in the conserved stem-loop structure [23]. The conserved structures of 3′ UTR of both GCXV and the JVG are also quite different from the terminal conserved structures of 3′ UTR of the *Flavivirus* genus. The same can be said about conserved structures in 5′ UTR. Comparison of those structures predicted by LocARNA is presented on Figure 5. Such differences may imply significant differences in the replication processes of these three virus groups. These UTR differences may be also used in the future as one of the markers for taxonomic differentiation among the JVG, the Guaico Culex virus group, and the *Flavivirus* genus.

Many segmented viruses have highly conserved sequences on terminal 5′ UTR nucleotides [23,26,28]. In the case of the JVG, current data are ambiguous. In all our sequences, both ALSV and YGTV shared 100% of the conserved 5′-AGUUAA-3′ sequence at the terminal 5′ end. This is consistent with data reported for JMTV during its first isolation, where 5′-GUUWA-3′ consensus was found in all the segments [3]. The terminal 5′ UTR 5′-AYAUGGG-3′ was reported in the first described ALSV genome [7]. However, for two segments, 5′-AGUUAA-3′ could be located in the alignment after 5′-AYAUGGG-3′ (Appendix A). Moreover, the data from GenBank represent extremely divergent terminal 5′ UTRs, likely due to RACE not being performed in most of the cases. Overall, more accurate data are needed to confirm or deny the strict conservation of terminal 5′ UTR nucleotides in the JVG.

We also identified an A-rich region within the JVG 3′ UTRs. It varied in size depending on the strain and segment and could cause problems with oligodT adapters during 3′ RACE, resulting in truncated UTR sequences (Appendix A). Moreover, approximately 60% of the GenBank entries had truncated 3′ UTR sequences. In our opinion, there are two possible explanations for this situation. First, some JVG members indeed had a shortened version of 3′ UTR depending on strain and segment. Second, most of the studies conducted using 3′ UTR RACE have been based on a specific adapter targeting polyA sequences. Moreover, many commercial 3′ UTR RACE protocols advise performing bacterial cloning after RACE to obtain cleaner Sanger sequences. With JVG genomes containing an A-rich region, it is easy for a polyA-adapter to mis-anneal onto the A-rich region. In addition, with a bacterial-cloning approach, there is a high chance of obtaining a shortened consensus sequence. One piece of evidence for this is that in a study where RACE based on RNA circulation was performed, only full JMTV 3′ UTRs were recovered [3]. With the data we have now, it is hard to tell if 3’ UTR shortening was caused by the sequencing and assembly methods used, or if these data mark an important feature of JVG biology. In any case, we think that, in the light of our cloning data, RNA circularization methods [3] should be used to obtain reliable information on JVG 3′ UTRs.

According to the predicted RNA structures, the three most-conserved sequences found in JVG genomes were involved in the formation of RNA stems with an AAGU tetraloop. Overall, loop motifs are important for the formation and stability of an RNA structure. Tetraloops can also function as recognition sites for proteins. There are six major classes of tetraloops [38]. While the AAGU tetraloop does not belong to any of the major classes, it has similarities with AGNN tetraloop structures. However, in the case of the AAGU tetraloop, the uracil is extruded from the tetraloop structure and does not form a non-canonical base pair with adenine, as seen in AGNN tetraloops [39].

Conserved regions in the UTR take part in viral replication, translation, packaging, and some other functions [24,25,26,27,28]. In the closest well-studied group—the *Flavivirus* genus—a conserved 5’ UTR Y-shaped structure plays a major role in the initiation of virus replication, with several other conserved RNA elements involved in translation and sfRNA formation [24,29,30]. If our bioinformatics predictions are correct, we may speculate that the AAGU tetraloop is necessary for JVG replication or for correct genome encapsidation.

The data presented here may be helpful for the construction of full-genome molecular copies of JMTV, ALSV, and YGTV. JVG members are widely distributed [8,9,15,16,17,18] and are potential human pathogens [7,17]; conserved regions can be utilized for the construction of PCR assays, allowing for a quick and reliable identification method for all members of the JVG, which is currently lacking. In addition, further studies of JVG molecular biology are needed to determine the role of the described conserved sequences. Here, we predicted RNA structures using bioinformatics software based on UTR alignments. However, in vitro RNA structure probing is needed to confirm our predictions and provide more precise information.

## 5. Conclusions

In this work, the 5′ and 3′ UTRs of several ALSV and YGTV strains were sequenced. An analysis of the obtained sequences revealed several highly conserved regions in both the 5′ and 3′ UTRs. RNA-folding programs predicted that conserved regions were likely to be involved in the formation of conserved stem-loop structures characterized by the presence of AAGU tetraloops. Sequences obtained from GenBank hinted that similar sequences and RNA structures could also be present in JMTV.

## Figures and Tables

**Figure 1 viruses-15-00971-f001:**
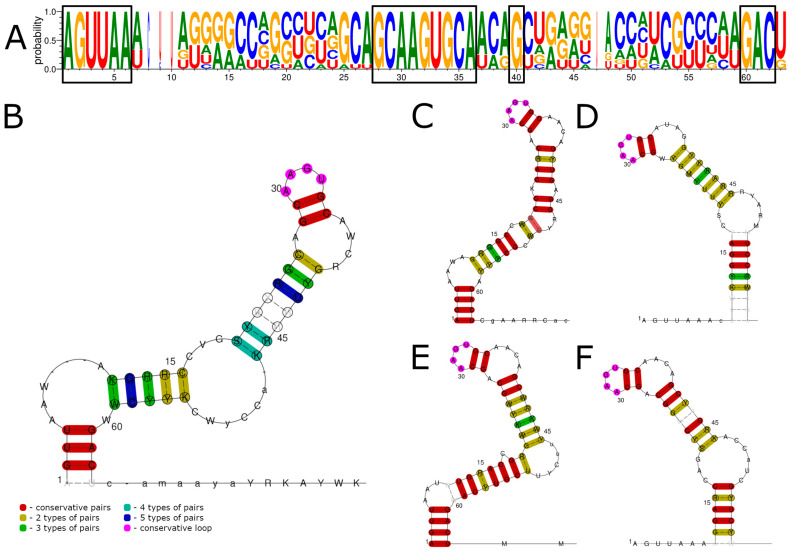
Consensus sequences and conserved RNA structures of 5′ UTRs of ALSV and YGTV studied in this work predicted using LocARNA. (**A**) Consensus sequences of all segments of the 5′ UTRs for ALSV and YGTV. (**B**) Conserved RNA structures of all segments of the ALSV and YGTV 5′ UTRs. Conserved RNA structures of segments one (**C**), two (**D**), three (**E**), and four (**F**). Conserved nucleotides in the consensus sequence are marked by black boxes. Nucleotide enumeration in (**B**–**F**) follows the enumeration in the alignment (**A**).

**Figure 2 viruses-15-00971-f002:**
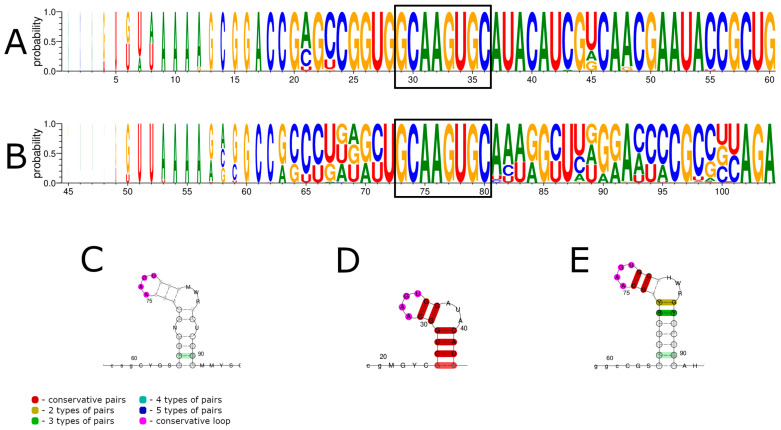
Consensus sequences and structures of JMTV 5′ UTRs predicted by LocARNA. (**A**) Consensus sequences of JMTV segment two 5′ UTRs. (**B**) Consensus sequences of JMTV 5′ UTR segments one, three, and four. (**C**) Consensus structures of all segments of JMTV 5′ UTRs. (**D**) Consensus structures of JMTV segment two 5′ UTRs. (**E**) Consensus structures of JMTV 5′ UTR segments one, three, and four. Conserved nucleotides in the consensus sequence are marked by black boxes. Nucleotide enumeration in (**C**–**E**) follows the enumeration in the alignment (**A**).

**Figure 3 viruses-15-00971-f003:**
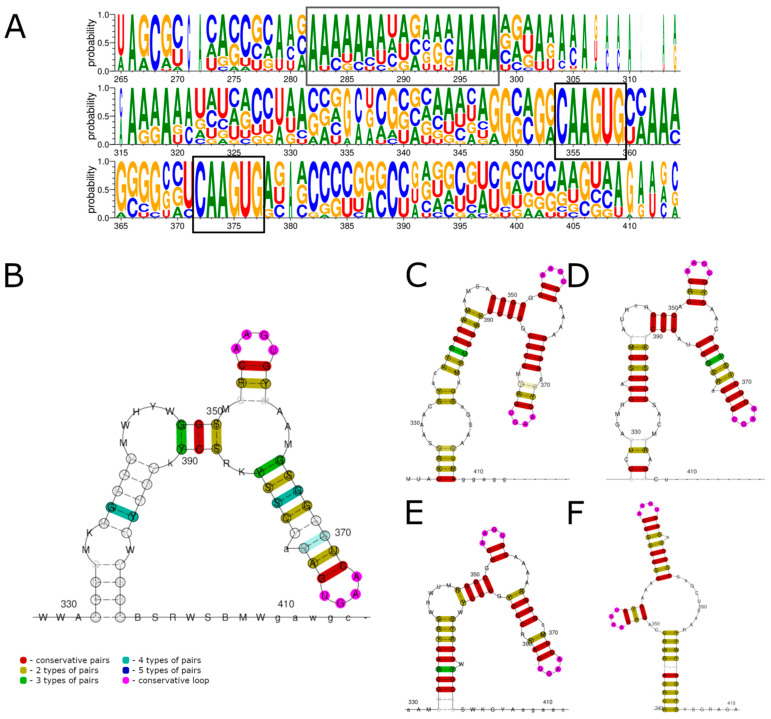
Consensus sequences and conserved RNA structures of 3′ UTRs of ALSV and YGTV studied in this work predicted using LocARNA. (**A**) Consensus sequences of all segments of ALSV and YGTV 3′ UTRs. (**B**) Conserved RNA structures of all segments of ALSV and YGTV 3′ UTRs. Conserved RNA structures of segments one (**C**), two (**D**), three (**E**), and four (**F**). Conserved nucleotides in the consensus sequences are marked by black boxes. Nucleotide enumeration in (**B**–**F**) follows the enumeration in the alignment (**A**).

**Figure 4 viruses-15-00971-f004:**
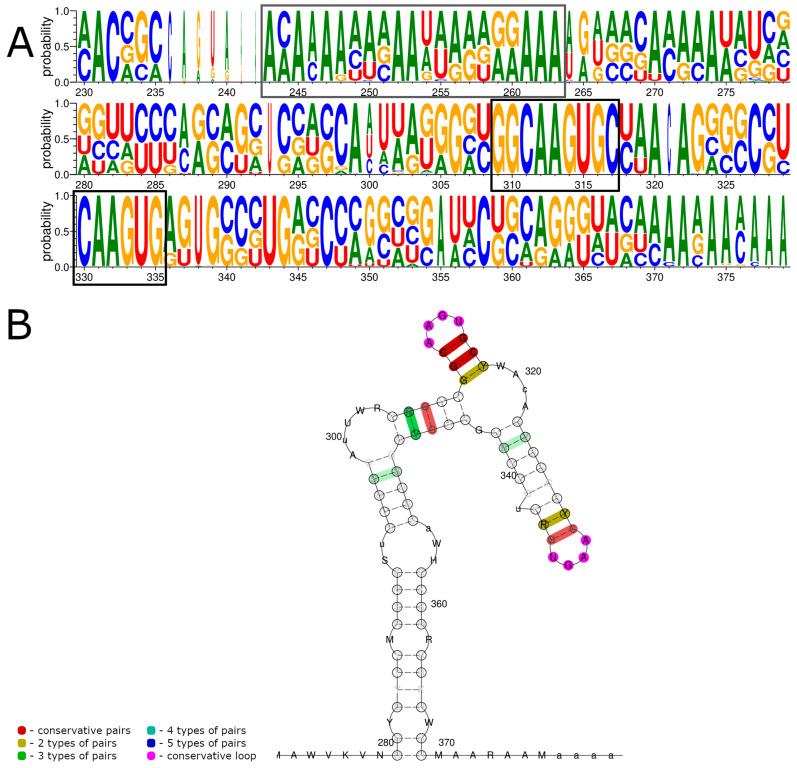
Consensus sequences of JMTV 3′ UTRs (**A**) and consensus structures predicted by LocARNA (**B**). Conserved nucleotides in the consensus sequences are marked by black boxes. Nucleotide enumeration in (**B**) follows the enumeration in the alignment (**A**).

**Figure 5 viruses-15-00971-f005:**
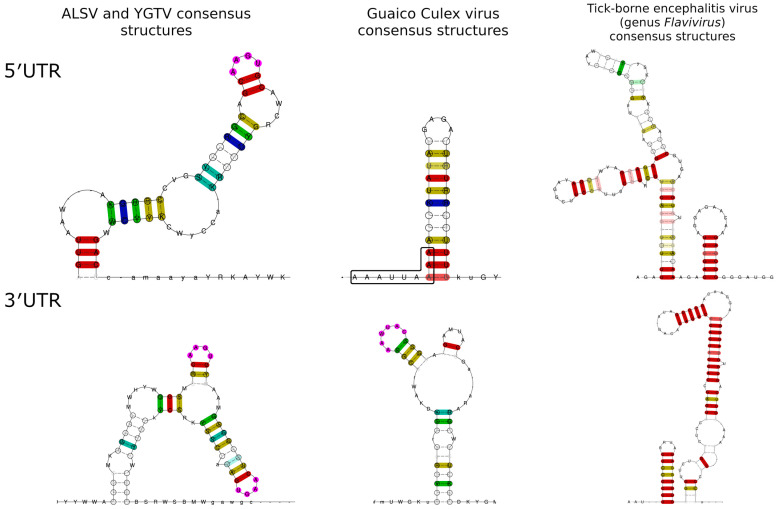
Comparison of the predicted secondary structures of the ALSV and YGTV and GCXV and tick-borne encephalitis virus (genus *Flavivirus*). Conserved 5′ UTR nucleotides of the GCXV are marked by black box. For the construction of the GCXV 3′ UTR structure, only segment 1–4 sequences were used. Conserved 3′ UTR loop of the GCXV is marked in purple. For the construction of TBEV 3′ UTR structure, 109 terminal nucleotides were used.

## Data Availability

All obtained sequences were deposited in GenBank. Entries ON448346–ON448374, OQ789403-OQ789442, and OQ789396-OQ789399 were deposited, and entries MN648770.2–MN648777.2, MW525314.2–MW525317.2, and MW525322.2–MW525325.2 were updated.

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
