# Peer review of "Conserved Sequences in the 5′ and 3′ Untranslated Regions of Jingmenvirus Group Representatives"

_viruses, 2023, doi:10.3390/v15040971_

Round 1

Reviewer 1 Report

Comments to Authors

The manuscript entitled “Conserved sequences in the 5′ and 3′ untranslated regions of Jingmenvirus group representatives” by Alexander G. Litov et al., performs a structural and sequential comprehensive analysis of the UTR-5 'and UTR-3' regions of Jingmenvirus group. Moreover, this bioinformatic analysis is carried out not only on database sequences, but also on novel sequences obtained by Race.

In my view, the manuscript presents a good structure and development. In addition, the manuscript analyzes and characterizes interesting regulatory genomic regions of Jingmenvirus group to increase the knowledge of its molecular biology. The characterization of these viral genomes (JVG Members) and the understanding of the functionality of their UTR is important since they are widely distributed and are potential human pathogens. So, the results presented would be useful for researchers in areas such as Virology, Biotechnology, Basic and Biochemistry Science. Therefore, it could be Accept after minor revision.

Minor revision:

- LINE 74, It would be more useful for the reader, if in addition to the names of the strains the GenBank IDs are indicated. It can be done in this section or at the end of the manuscript (Data Availability Statement), where IDs are indicated (but without the names of the Strains).

- LINE 88, It is suggested to replace "Cultural supernates" with culture supernatants.

- LINE 128, “In sequences where clones were taken, after the assembly of individual clones,”. Rewrite the sentence because it is not fully understood. It would be: "In the sequences where several clones were taken, after the assembly of individual clones," or "In cases where several clones were taken, after the assembly of individual clones,".

- LINE 269, Likewise, that same sequence presents another change in a nearby region (C for T, position 106 of the alignment in Figure S6), which could allow the formation of the same structure despite the nucleotide change (there are two mutual mutations, allowing the formation of a bond structure preservation). Many times in these cases, structural conservation is more important than sequential conservation, so if changes that maintain the structure occur, there are no changes in the function of that region.

- LINES 301-311, To further discuss the concept developed in this paragraph, about the structural differences between GCXV and JVG, it would be very useful to have an image of UTR structures of both, GCXV and the Flavivirus genus. It is not necessary that it be a structure predicted with the consensus of all the flavivirus sequences, it would be enough with a representative structure (of a single genome) to visually show the differences. Furthermore, it would be interesting to gather in a single figure the structures of the UTR-5' and UTR-3' of JVG, GCXV and Flavivirus; marking structural differences.

Author Response

Reviewer report:

The manuscript entitled “Conserved sequences in the 5′ and 3′ untranslated regions of Jingmenvirus group representatives” by Alexander G. Litov et al., performs a structural and sequential comprehensive analysis of the UTR-5 'and UTR-3' regions of Jingmenvirus group. Moreover, this bioinformatic analysis is carried out not only on database sequences, but also on novel sequences obtained by Race.

In my view, the manuscript presents a good structure and development. In addition, the manuscript analyzes and characterizes interesting regulatory genomic regions of Jingmenvirus group to increase the knowledge of its molecular biology. The characterization of these viral genomes (JVG Members) and the understanding of the functionality of their UTR is important since they are widely distributed and are potential human pathogens. So, the results presented would be useful for researchers in areas such as Virology, Biotechnology, Basic and Biochemistry Science. Therefore, it could be Accept after minor revision.

Reviewer suggestion:

- LINE 74, It would be more useful for the reader, if in addition to the names of the strains the GenBank IDs are indicated. It can be done in this section or at the end of the manuscript (Data Availability Statement), where IDs are indicated (but without the names of the Strains).

Answer:

Thank you for the suggestion. We tried to make this correction, but in the end it seemed to us that addition of the all of the GenBank IDs within the body of the manuscript will make it quite bulky and not convenient to read and navigate. 

Reviewer suggestion:

- LINE 88, It is suggested to replace "Cultural supernates" with culture supernatants.

Answer:

Text was replaced.

Reviewer suggestion:

- LINE 128, “In sequences where clones were taken, after the assembly of individual clones,”. Rewrite the sentence because it is not fully understood. It would be: "In the sequences where several clones were taken, after the assembly of individual clones," or "In cases where several clones were taken, after the assembly of individual clones,".

Answer:

Text was adjusted according with your second suggested variant.

Reviewer suggestion:

- LINE 269, Likewise, that same sequence presents another change in a nearby region (C for T, position 106 of the alignment in Figure S6), which could allow the formation of the same structure despite the nucleotide change (there are two mutual mutations, allowing the formation of a bond structure preservation). Many times in these cases, structural conservation is more important than sequential conservation, so if changes that maintain the structure occur, there are no changes in the function of that region.

Answer:

In a 5′-CAAGUG-3′ region discussed in the text (Figure S6, positions 99-104), C(99) is paired with G(104). In the discussed MH688534 sequence, C(99) is substituted by A(99). Thus true compensating substitution would be a G(104) into T(104).

The situation you pointed out would result in similar structure in terms of stem nucleotide pairing, but RNA would have a 5′-AAGUGA-3′ loop instead of 5′-AAGU-3′ that seems to be conservative. That’s why, we did not mentioned this as a case of the structural conservation.

Reviewer suggestion:

- LINES 301-311, To further discuss the concept developed in this paragraph, about the structural differences between GCXV and JVG, it would be very useful to have an image of UTR structures of both, GCXV and the Flavivirus genus. It is not necessary that it be a structure predicted with the consensus of all the flavivirus sequences, it would be enough with a representative structure (of a single genome) to visually show the differences. Furthermore, it would be interesting to gather in a single figure the structures of the UTR-5' and UTR-3' of JVG, GCXV and Flavivirus; marking structural differences.

Answer:

Thank you for the suggestion. We calculated an additional structures presenting 5'UTR and 3'UTR of the GCXV and tick-borne encephalitis virus using similar methodology. We added a single figure where all calculated structures are presented, with discussed elements of structures highlighted.

Matherials and Methods section was adjusted accordingly to describe the protocol used for TBEV and GCXV.

Reviewer 2 Report

Litov et al. present a manuscript on the "Conserved sequences in the 5′ and 3′ untranslated regions of 2 Jingmenvirus group representatives". In their work, complete untranslated regions (UTRs) of four strains of ALSV and eight strains of 13 YGTV were obtained, and these sequences were used to identify homologous and orthologous sequences in the database. Their analyses identified several regions within viral UTRs that were highly conserved for all the segments and viruses, and their bioinformatics predictions suggested that the UTR of all the segments of YGTV, ALSV, and JMTV could form similar RNA structures. The most notable feature of these structures was a stable 17 stem-loop with one (5′UTR) or two (3′UTR) AAGU tetraloops on the end of a hairpin.

The work described fit well with the basal in silico protocols used for comparative sequence analyses, and the data described is quite basic. Moreover, no functional studies were performed, including mutagenesis, to disrupt secondary structures on the conserved UTRs; the work is purely descriptive, and not much new is learnt for the data acquired to justify presentation in a full-paper format. The Reviewer strongly suggests that the work can be submitted in the form of a very SHORT COMMUNICATION to highlight the sequence information, and perhaps phylogeny. Much of the work can be referenced in "supplememtal" form.

RECOMMENDATION:

1. REJECT

2. Consider reformating the manuscript to a concise Short Communication for another round of peer-review.

Author Response

Thank you for the review.

We agree that, in order to obtain accurate information on RNA structures, approaches allowing direct pairing deciphering (such as selective 2'- hydroxyl acylation and primer extension) are needed. And in order to report on the functional importance of said RNA structure elements, mutagenesis studies needed to be performed. We are currently working on setting up appropriate methodical background in our lab, but it is impossible to be done from zero quickly.

At the same time, viruses analyzed here are quite divergent (about 20% amino acid divergence in the polymerase protein), collected across vast distances geographically and dataset is quite big (more than 500 sequences), making our conservation data reliable. Recent bioinformatics approaches performed on a large datasets proved to be very effective in highlighting points of interests in the viral genomes, and with predictions confirmed by structural and\or mutagenesis studies (see, for example Lulla V, et al. An upstream protein-coding region in enteroviruses modulates virus infection in gut epithelial cells. Nat Microbiol. 2019;4(2):280-292).

Moreover, Jingmenvirus group is discovered quite recently, and a vast majority of the sequencing does not use any special approaches in order to obtain an accurate UTR sequences. In our article we report on common conservation patterns within UTRs of all four segments of all known jingmenviruses, that can be used in further studies for the construction of the full-genome molecular copies, construction of PCR assays for Jingmenvirus group detection, oligonucleotide design for novel members of Jingmenvirus group and so on. For example in one article on the topic (Kobayashi, D.; et al. Detection of Jingmenviruses in Japan with Evidence of Vertical Transmission in Ticks. Viruses 2021, 13, 2547) it is said that authors were unable to design oligonucleotides for 5 UTR of the novel jingmenvirus because “sequence and structure of the 5′ termini of each genome segment of known jingmenviruses remains to be determined”. Thus knowledge of the conservation region is important to the scientists in the field. At the same time, structure prediction data can bring the new point of attention for future possible studies.

Reviewer 3 Report

This is a concise and well written paper that describes sequence analysis of UTR of the genome of a number of viruses that are members of the relatively less understood Jingmenvirus group.  These arboviruses are becoming increasing important as they infect a wide range of species and are possible agents of human disease.  The analyses revealed a number of highly conserved regions that formed distinctive RNA hairpin structures with one or two stable stem and tetra loops.  The implications for these structures in virus replication and translation are discussed and the conclusions drawn are appropriate based on the data presented, which lead into potential future areas of further study.  The bioinformatic data analyses and the derived structural data have been clearly presented in the figures in the paper or in the supplementary data, with an appropriate balance between the two.

Author Response

We would like to thank reviewer for such a positive report.

Round 2

Reviewer 2 Report

The authors have adequately addressed my previous concerns.